# Actional Mechanisms of Active Ingredients in Functional Food Adlay for Human Health

**DOI:** 10.3390/molecules27154808

**Published:** 2022-07-27

**Authors:** Yawen Zeng, Jiazhen Yang, Jia Chen, Xiaoying Pu, Xia Li, Xiaomeng Yang, Li’e Yang, Yumei Ding, Mingying Nong, Shibao Zhang, Jinbao He

**Affiliations:** 1Biotechnology and Germplasm Resources Institute, Yunnan Academy of Agricultural Sciences/Agricultural Biotechnology Key Laboratory of Yunnan Province, Kunming 650205, China; chenjia@yaas.org.cn (J.C.); puxiaoying@163.com (X.P.); lixia_napus@163.com (X.L.); yxm89ccf@126.com (X.Y.); yangyanglie@163.com (L.Y.); dym@yaas.org.cn (Y.D.); 2Key Laboratory of the Southwestern Crop Gene Resources and Germplasm Innovation, Ministry of Agriculture, Kunming 650205, China; yangjiazhen415@163.com; 3Wenshan Academy of Agricultural Sciences, Wenshan 663099, China; nmyynws@163.com (M.N.); zsbynws@163.com (S.Z.)

**Keywords:** adlay, medicine and food homologous, functional ingredient, anti-cancer, anti-obesity, anti-inflammatory, anti-diabetic, anti-oxidant, actional mechanisms

## Abstract

Medicinal and food homologous adlay (*Coix lachryma-jobi* L. var. *ma-yuen* Stapf) plays an important role in natural products promoting human health. We demonstrated the systematic actional mechanism of functional ingredients in adlay to promote human health, based on the PubMed, CNKI, Google, and ISI Web of Science databases from 1988 to 2022. Adlay and its extracts are rich in 30 ingredients with more than 20 health effects based on human and animal or cell cultures: they are anti-cancer, anti-inflammation, anti-obesity, liver protective, anti-virus, gastroprotective, cardiovascular protective, anti-hypertension, heart disease preventive, melanogenesis inhibiting, anti-allergy, endocrine regulating, anti-diabetes, anti-cachexia, osteoporosis preventive, analgesic, neuroprotecting, suitable for the treatment of gout arthritis, life extending, anti-fungi, and detoxifying effects. Function components with anti-oxidants are rich in adlay. These results support the notion that adlay seeds may be one of the best functional foods and further reveal the action mechanism of six major functional ingredients (oils, polysaccharides, phenols, phytosterols, coixol, and resistant starch) for combating diseases. This review paper not only reveals the action mechanisms of adding adlay to the diet to overcome 17 human diseases, but also provides a scientific basis for the development of functional foods and drugs for the treatment of human diseases.

## 1. Introduction

Human health is dependent on a plentiful supply of functional foods [1], especially functional rice, barley, and adlay. High-resistant starch rice can be stored long term and quantified, is considered a safe and a maximally functional food, and has a remarkable effect on controlling postprandial blood sugar, reducing its complications and the required insulin dosage, which could help solve the global medical problem of hypoglycemia among diabetics [2]. Functional rice and barley have become an important part of functional food crops, while functional food crops lead the development of functional agriculture with plateau characteristics [3]. Barley plays an important role in health and was important to civilization during the human migration from Africa to Asia, and later to Eurasia [4]. Coix is a grass crop domesticated in the Neolithic era that diverged from sorghum ~10.41 million years ago [5]. 

Adlay seeds (yì yĭ “soft-shelled job’s tears” or *Coicis Semen*; the seeds of *Coix lachryma-jobi* L. var. *ma-yuen* Stapf) is a grass crop that has long been used in traditional Chinese medicine, and is considered a functional food in China and the East, as well as Southeast Asia, for the treatment of warts, chapped skin, rheumatism, neuralgia, and inflammation and in stomachic, diuretic, anodynic, anti-pholgistic, anti-spasmodic, and neoplastic diseases [6]. Adlay or Chinese pearl barley belongs to the family of Gramineae herbaceous plants. Both adlay and its by-products (roots, stems, and leaves) are homologous substances of medicine and food. Coix and barley are widely planted all over the world and are known as “the king of Gramineae plants” [7]. 

There is some epidemiological evidence from human trials and systematic reviews on the use of adlay. The bioactive compounds from adlay essential oil hinder the interaction of SARS-CoV-2 with the ACE2 receptor, which suppresses viral replication and may prevent COVID-19 continuing as an ongoing pandemic in 2022 [8]. The essential oil of roots and adlay might be useful in the bacterial meningitis treatment, as it contains steroids, carotenoids, tannins, alkaloid salts, reducing compounds, flavonoids, anthracenoids, coumarin derivatives, cardenoid, anthocyanins, and saponins [9]. Adlay could promote the spontaneous regression of viral skin infections, and its oil can increase the peripheral CD4+ lymphocytes, according to a comparison of intervention and no intervention in two groups of healthy adult males [10]. A significant decrease in the systolic blood pressure among people who consumed 60 g of dehulled adlay daily for a six-week period has also been observed [11].

Adlay contains oils, polysaccharides, phenols, coixol, phytosterols, resistant starch, flavonoids, lactams, fatty acids and esters, proteins, vitamins, triterpenes, alkaloids, fibers, and other compounds, which can treat many chronic diseases (cancers, hypertension, fatty liver, hyperlipidemia, anti-obesity, anti-inflammation, and rheumatoid arthritis; it can also enhance immunity, regulate intestinal flora, stimulate reproductive hormones, promote uterine contraction, modulate gut microbiota, and act as an anti-oxidant). In traditional Chinese medicine, it is used to remove dampness and for diuresis [6,12,13]. Therefore, these effects suggest that the health effects of adlay are the result of the long-term coevolution of human *Coicis Semen* as food; however, the lack of a systematic understandings of functional ingredients in adlay remains an issue. This paper reviews the nutritional and functional ingredients, pharmacological actions, action pathways, molecular mechanisms and anti-oxidants of six functional ingredients in adlay to combat diseases. 

## 2. Functional Ingredients in Adlay

There are about 30 nutritional and functional ingredients in the different tissues (seed, coat, hull, root, stem, and leaves) of adlay [14] (see Table 1). Higher contents of proteins (19.33%) and polysaccharides (2.26%) were determined in adlay grown in Guizhou. Adlay root had the highest content of V_B1_ (0.61 mg/kg) and V_B2_ (2.96 mg/kg) and eoixol (1.26 mg/g); however, the contents of unsaturated fatty acids in adlay coat, stem, and leaves were more than 70% [14] (Table 1). Eight functional ingredient contents (protein, polysaccharides, resistant starch, Zn, P, amino acids, palmitate, and flavonoids) in adlay seeds were at the highest level among six tissues (seeds, coat, hull, root, stem, and leaves) of adlay (see Table 1). Eleven functional ingredient contents (crude fat, five vitamins, Fe, Se, palm-linoleic acid, eoixol, and total phenol) contents in adlay root were at the highest level among six tissues (seeds, coat, hull, root, stem, and leaves) of adlay (see Table 1). Nine functional ingredient contents (V_B6_, folciate acid, K, Ca, Na, Mg, oleic acid, linoleic acid, and decoanic acid) in adlay leaves had the highest levels among six tissues (seeds, coat, hull, root, stem, and leaves) of adlay (see Table 1) [14]. These findings show that adlay seeds and leaves are the ideal raw materials for human functional foods and livestock functional forages.

Adlay has been found to significantly reduce nausea and vomiting in chemotherapy [15]. Six ingredients (coniferyl alcohol, syringic acid, syringaresinol, 4-ketopinoresinol, ferulic acid, and mayuenolide) of adlay hull showed very strong anti-oxidant activity [16]. Six ingredients (*p*-hydroxybenzaldehyde, syringaldehyde, vanillin, sinapaldehyde, coixol, and *trans-*coniferylaldehyde) of acetone extract from adlay hull show strong anti-mutagenic activity, especially *trans-*coniferylaldehyde as an anti-cancer chemoprevention, which can activate five kinase signals [17]. Adlay coixol can regulate gene expression and secretion of mucin by acting on airway epithelial cells [18].

The total protein contents were from 15.63% to 25.74%, and crude fat contents were from 5.05% to 7.14%. Alcohol extract (5.85–7.27%) of Coix at an altitude of 550–1550 m in Xishuangbanna showed particularly strong anti-tumor activity [19], but the fat and protein levels in adlay from Wenyi2 in the Wenshan prefecture of Yunnan province were 6.3% and 15.25%. The highest levels of phenolic acid and flavonoid in subfraction 3 of defatted adlay seed were 67.28 mg/g and 41.11 mg/g, *p*-coumaric acid showed the highest oxygen radical absorbance capacity, while quercetin exhibited the highest peroxyl radical scavenging capacity and cellular anti-oxidant activity [20]. There were notable differences in contents of starch (65.54 ± 3.28%), fat (7.82 ± 0.60%), protein (18.18 ± 1.36%), and amino acid (17.38 ± 1.21%) among 86 adlay landraces [21] (Table 2). The highest triolein content (1.04%) in adlay was found in Zhejiang province and the highest coixol content (3.47 mg/g) in adlay leaves was found in Shanghai [22]. Adlay bran had higher contents of policosanols (246 mg/kg), phytosterols (4733 mg/kg), and oleamide (45.8 mg/kg) [23]. Total phenolic (2.71 ± 0.02 mg/g), coixol (59.70 ± 0.01 mg/g), and flavonoid (0.60 ± 0.02 mg/g) contents, as well as anti-oxidant capacity (453.93 μg/mL) of adlay sprouts, were highest when they were developed as natural anti-oxidants [24]. A 60 h germination brought about a 3.4-fold increase in GABA and 3.6-fold increase in coixol compared to ungerminated adlay seeds [25]. 

There are higher tetramethylpyrazine, fibrinolytic enzymes, and functional ingredients (phenolics, GABA, triterpenes, flavonoids, and coixenolide) in fermented adlay products, making them more suitable for human consumption [30]. The compound structures [9-hydroxy-(10*E*)-octadecenoic acid, 13-hydroxy-(9*E*,11*E*)- octadecadienoic acid, 9-hydroxy-(10*E*,12*E*)-octadecadienoic acid(9-*E*,*E*-HODE), 10- hydroxy-(8*E*)-octadecenoic acid, 8-hydroxy-(9*E*)-octadecenoic acid, 11-hydroxy-(9*Z*)- octadecenoic acid,9-*E*,*E*-HODE] from adlay seed extracts with acetone and 70% ethanol can act as peroxisome proliferator-activated receptor gamma agonists [31]. All ethyl acetate fractions contain high contents of carotenoid and tannin, whereas the hexane soluble fraction of the roasted hull of Laos Black Loei Job’s tears showed the highest linoleic acid content of 8.09 ± 0.74% *w*/*w* [32]. Hydroxy-(9*Z*)-octadecenoic acid was isolated from adlay seeds, and hydroxylation of monounsaturated fatty acids can silence polyunsaturated fatty acids ligands to activate polyunsaturated fatty acids agonists [33]. Coixins from adlay obtained α-, β-, and ϒ-coixin, especially α-coixins, which were composed of four polypeptides, and C1 and C2 α-coixins corresponded to 80% of the total coix prolamins, while anti-serum against ϒ-coixin showed a strong cross-reaction with ϒ-zein [34]. 

## 3. Action Pathway against Disease of Adlay

Adlay is among the most widely recognized Chinese herbal medicines for its remedial effects against inflammation, endocrine system dysfunction, warts, chapped skin, neuralgia, and rheumatism [35,36]. Adlay has shown a good treatment effect in many difficult diseases. This review revealed more than 20 health effects based on human, animal, and cell culture experiments (see Table 3 and Table 4, Figure 1). The mean divergence times between Coix and Sorghum is 9.3~12.4 million years ago; these genera diverged from a common progenitor, and underwent evolution for about 50 million years [37]. The health effects of functional components in adlay are similar to barley as a result of long-term continuous evolution of early hominids (fruits/vegetables and leaves rich in polyphenols), Neanderthals (mushrooms and nuts rich in polysaccharides, phytosterols), and *Homo sapiens* (grasses and seeds rich in resistant starch) [38].

### 3.1. Anti-Cancer Effects

Table 3 shows the action pathways and functional ingredients of adlay in the treatment of thirteen cancer types. The functional ingredients of adlay with anti-cancer mainly include oils (triglycerides, fatty acid), polysaccharides, phenols, coixol, phytosterol, resistant starch, coixenolide, and flavonoids (see Table 3 and Figure 2). Sepich-Poore et al. [83] revealed the causal and complicit roles of the immuno-oncology -microbiome axis for microbes in cancer. Adlay sprout extract not only inhibits metastatic activity and adhesion of colon cancer cells as well as tube formation by HUVECs via repression of the ERK1/2 and AKT pathways [48], but also causes cell cycle arrest and apoptotic cell death through the inactivation of the PI3K/AKT pathway in HeLa cells [35], suggesting it is a good anti-cancer (colon and cervical) therapeutic agent. Adlay seed oil and tripterine co-loaded microemulsions with a transferrin modification enhance tumor-targeting, facilitate deep penetration of drugs, decreased concentrations in serum, and efficient treatment for cervical cancer by regulating bax/bcl-2 and the activating caspase-3 pathway [40]. The steamed whole Thai Black adlay extract showed the highest anti-proliferative activity in human mouth epidermal cancer cell at an IC50 of 43.61 ± 0.76 μg/mL, whereas the roasted whole Laos White adlay extract showed the highest apoptotic activity in cervical adenocarcinoma at 21.52 ± 1.50% [32]. Phytosterols (stigmasterol and β-sitosterol) from the adlay hull fraction can be used as a natural medicine in the treatment of uterine leiomyoma growth [54]. Four compounds have an anti-proliferative effect (polyphenols (protocatechuic acid, *p*-hydroxybenzaldehyde, caffeic acid), flavonoids, phytosterols, and fatty acids) in adlay seed fractions (ethyl acetate) against endometrial cancer cells [41].

Injectable adlay extract induces apoptosis in the hepatoma cancer cells by regulating the expression of Caspase-8 [84]. The synergy of a nanostructured lipid carrier with naringin in coix seed oil showed significantly enhanced anti-tumor efficacy in a xenograft model of liver cancer [85]. An octanoyl galactose ester-modified microemulsion system, self-assembled by coix seed components, is a highly effective and safe anti-cancer (hepatoma) drug delivery system; its half-maximal inhibitory concentration against HepG2 cells was 46.5 ± 2.4 μg/mL [42]. Triterpene-loaded microemulsions using adlay seed can treat lung cancer in an efficient drug delivery system, which showed the strong anti-proliferative effect on human lung tumor cells [56]. Adlay seed extract could increase the efficacy of gemcitabine therapy and mitigate its upregulation of ABCB1 and ABCG2 proteins in pancreatic cancer cells, due to the alteration of the ABC transporter-mediated drug efflux function [86].

The potent anti-proliferative activity against HT-29 cells of liposomes entrapped with adlay with high physicochemical stability could enable it to be developed as a novel anti-cancer drug for humans [87,88]. An EtOAc extract of adlay showed strong anti-tumor activity and was a natural anti-tumor bioactive agent, which produced triolein reached at a rate of 2.536 ± 0.006 mg/g dry weight of mycelium [89]. Compared to the raw seed, the tocols, γ-oryzanol, and coixenolide contents in the fermented coix seed from *M. purpureus* increased approximately 4, 25, and 2 times, respectively, while γ-tocotrienol and γ-oryzanol reached 72.5 and 655.0 μg/g, respectively, which increased the free and readily bioavailable lipophilic anti-oxidants and anti-cancer activity [90]. Four free fatty acids of adlay seeds possessing anti-tumor activity are palmitic, stearic, oleic, and linoleic acids [91]. The anti-tumor (20*R*)-22*E*-24-ethylcholesta-4,22-dien-3-one was isolated from the stems and leaves of adlay, and had pro-apoptotic effects on three types of tumor cells, as well as having an inhibitory effect on DNA topoisomerase I [69]. The proportion of palmitic acid and linoleic acid to oleic acid displayed a highly significant positive correlation with the inhibition rates of adlay seed oil for cancer cells [92]. Five active compounds (coixspirolactam (A,B,C), coixlactam, and methyl dioxindole-3-acetate) from adlay bran have anti-proliferative effects against A549, HT-29, and COLO 205 cells; their anti-cancer activities have IC50 values between 28.6 and 72.6 microg/mL [59]. 

### 3.2. Anti-Inflammation Effects

The functional ingredients of adlay that have anti-inflammation effects mainly include polyphenols, coixol, polysaccharides, eriodictyol, ceramide, and *p*-coumaric acid (see Table 4 and Figure 2). Adlay promotes transfer signals from the cell membrane into the cytosol and nucleus, triggering gene expression and changes in cell proliferation and apoptosis or DNA repair; however, cell proliferation is inhibited through downregulated COX-2 by polyphenols and polysaccharides. Polysaccharides can reduce mitochondrial membrane strength and induce Caspase-3 and 9-mediated apoptosis [60]. The eriodictyol and ceramide and *p*-coumaric acid of ethanol extract in adlay seed hulls has anti-inflammatory properties through increasing cellular production of nitric oxide and prostaglandin E2, induced by lipopolysaccharides by downregulating inducible nitric oxide synthase and cyclooxygenase 2 expression [61]. The methanol extract of adlay seeds shows anti-inflammatory properties, which involves an inhibition of nitric oxide and superoxide production by activated macrophages [93]. The AcOEt fraction of unhulled adlays has anti-inflammatory activity in RAW 264.7 cells [94]. Adlay seed oil markedly prevents obesity and improves systemic inflammation [62]. The anti-inflammatory activities of six benzoxazinoids isolated from adlay roots, especially concanavalin A, can be used in the first-stage screening of active compounds, and the free hydroxyl group at the 2-position in the benzoxazinone skeleton is important for the expression of inhibitory activity [95]. The ethanolic extract of adlay testa has anti-oxidative and anti-inflammatory activities, which are related to its phenolics, such as chlorogenic acid, 2-*O*-β-glucopyranosyl-7-methoxy-4((2)*H*)-benzoxazin-3-one, caffeic acid, ferulic acid, vanillic acid, and *p*-coumaric acid [96]. 

### 3.3. Anti-Obesity Effects

Obesity is a major public health concern worldwide, with a rising prevalence. The functional ingredients of adlay with anti-obesity effects mainly include phytosterol, resistant starch, polyphenols, and dietary fiber (see Table 4 and Figure 2). Daily intake of dehulled adlay had beneficial effects in blood pressure management, significantly decreasing in the systolic blood pressure, and this effect is more evident in participants with high baseline systolic blood pressure based on 23 participants with overweight and obesity consuming 60 g of dehulled adlay daily for a six-week experimental period [11]. Dietary intake of 60 g dehulled adlay per day may reduce body fat mass, inflammation, cholesterol, triglycerides, plasma tumor necrosis factor alpha, interleukin-6, leptin, and malondialdehyde in overweight and obese people; however, participants with higher basal blood lipid levels exhibited enhanced lipid lowering effects [97]. Adlay seed is a prebiotic drug to prevent obesity-related diseases based on glycerolipid metabolism, biosynthesis of unsaturated fatty acids, sulfur reduction, and the glutathione transport system [62]. AMP-activated protein kinase in adipose differentiation and adlay ethanol extract targeted to adipocytes result in decreased expression levels of adiogenesis factors such as fatty acid synthase, sterol-regulatory-element-binding protein-1c, peroxisome proliferator-activated receptor γ, and CAATT/enhancer binding protein α, effectively improving the symptoms of metabolic syndrome [63]. Adlay seed crude extract modulates the expressions of leptin and TNF-alpha, and reduced body weight, food intake, fat size, adipose tissue mass, and serum hyperlipidemia in obese rats fed a high-fat diet [98]. A water extract of adlay seed can treat obesity through neuroendocrine modulation in the brain [99].

### 3.4. Protective Liver Effects

The functional ingredients of adlay that are protective to the liver mainly include oils, phytosterol, and proteins (see Table 4 and Figure 2). Lipid metabolism disorder is the key dysfunction in non-alcoholic fatty liver disease (NAFLD). Adlay seeds extracts can alleviate NAFLD, and related liver and metabolic diseases due to inhibiting liver lipogenesis and inducing β-oxidation of fatty acids [100]. Adlay seed oil had a therapeutic effect in NAFLD. The degradation mechanism of adenosine 5′-monophosphate (AMP)-activated protein kinase results in reduced SePP1/apoER2 expression, and thus, reduces lipid accumulation [101]. Adlay seed protein hydrolysates have significant liver protection effects, significantly improving alcohol metabolism in the liver by enhancing aspartate aminotransferase, alanine aminotransferase, serum tumor necrosis factor-α, and interleukin-β, as well as retarding lipid peroxidation by inhibiting malondialdehyde levels and increasing the activity of liver superoxide dismutase [102].

### 3.5. Anti-Virus Effects

Emerging viruses such as SARS-CoV-2, Ebola, Lassa, and avian influenza virus are global health concerns. Adlay extracts exhibit potent anti-Ebola viral activities [103]. Adlay seeds are currently used as a traditional medicine for the treatment of COVID-19 in China [104,105]. Adlay seeds not only increase peripheral cytotoxic lymphocytes and may be effective for viral infection by enhancing cytotoxic activity [64], but also played a role in the COVID-19 treatment as one of the recommended prescriptions by the General Office of the National Health Commission and Office of the State Administration of Chinese Medicine [106].

### 3.6. Cardiovascular Protection Effects

The functional ingredients of adlay with cardiovascular protection effects mainly include polyphenols and phytosterols (see Table 4 and Figure 2). Polyphenol extract of adlay significantly reduces serum and low-density lipoprotein cholesterol and significantly improves high-density lipoprotein cholesterol, which may make it a functional food for elevated cholesterol levels and ecological imbalance in intestinal microbes [65]. Polyphenols from adlay effectively reduce total serum cholesterol and low-density lipoprotein cholesterol and malondialdehyde, improve serum high-density lipoprotein cholesterol and anti-oxidant capacity, and support anti-oxidant capacity, catalase, and glutathione peroxidase activities in liver [37]. Adlay seeds have important nutritional value in terms of reducing the fibrinogen level and reducing fibrinolytic activity, which can affect the development of human atherosclerosis [107].

### 3.7. Gastroprotection Effects

The functional ingredients of adlay with gastroprotection effects mainly include phenolic acids, polysaccharides, caffeic acids, and chlorogenic acids (see Table 4 and Figure 2). The anti-oxidative-active phenolic acids and anti-ulcer activity in dehulled adlay promote partial gastric protection, and ethanol extracts (19 compounds) of adlay bran show better anti-proliferative activity, especially caffeic and chlorogenic acids, which significantly inhibit the growth of AGS cells [108]. Adlay seeds promote the growth of pigs after weaning by reducing the pH value of gastric juice, increase the density and length of gastrointestinal fluff, and regulate intestinal microbiota [66]. 

### 3.8. Anti-Hypertension Effects

The functional ingredients of adlay with anti-hypertension effects mainly include gluten, glutelin, and phytosterols (see Table 4 and Figure 2). Angiotensin inhibiting peptide GAAGGAF of adlay glutelin-converting has a significant anti-hypertensive effect; GAAGGAF at 15 mg/kg body weight in spontaneously hypertensive rat could reduce the systolic pressure by about 27.50 mmHg, and the blood pressure-lowering effect lasted for at least 8 h, making it a natural component for pharmaceuticals to combat hypertension and related diseases [67]. The polypeptides produced by gluten and alcohololytic protein hydrolysis in adlay have anti-hypertensive effects and inhibit angiotensin activity in vitro [6]. Adlay peptides involve a multi-regulatory mechanism that regulates blood pressure, especially high angiotensin I converting enzyme (ACE) inhibitory activity. Tetrapeptide VDMF significantly downregulates ACE, AngII type 1 receptor, and ACE2 mRNA expression, while upregulating *Mas* gene expression [68].

### 3.9. Preventative of Heart Disease

The functional ingredients of adlay that are preventative of heart disease mainly include oils (triglycerides) and vitamin E (see Table 4 and Figure 2). Adlay seed oil has blood lipid-reducing and anti-oxidant effects, and could be used to prevent chronic diseases, especially atherosclerosis and coronary artery disease [69]. Vitamin E in adlay prevents cancer and heart disease, and the effectiveness of assembled gene sets by the levels of prolamin and vitamin E biosynthesis-associated proteins in adlay has been confirmed [70].

### 3.10. Anti-Allergy Effects 

The functional ingredients of adlay with anti-allergy effects mainly include phenolic acids, flavone, *p*-coumaric acid, and hydroxyacetophenone (see Table 4 and Figure 2). An allergy is an immune dysfunction caused by degranulation from mast cells in the early phase and cytokine secretion in the late phase of the cell. Adlay bran extract (40.8 μg/mL) inhibits mast cell degranulation, reduces the release of histamines and cytokines, suppresses Akt production, and affects the signaling transduction of RBL-2H3 cells. Luteolin in six phenolic acids and one flavone exhibited the strongest inhibitory activity (1.5 μg/mL), thus revealing the anti-allergy mechanisms of adlay [71]. The ethanolic extracts (4-hydroxyacetophenone and *p*-coumaric acid) of adlay testa had an inhibitory effect on allergic reactions via the ERK signaling transduction of RBL-2H3 cells [72].

### 3.11. Melanogenesis Inhibition 

The functional ingredients of adlay that show melanogenesis inhibition mainly include oils and coixol (see Table 4 and Figure 2). Adlay bran oil reduces tyrosinase activity and melanin production. Melanoma cells and zebrafish have tyrosinase inhibitors and anti-hyperpigmentation agents [73]. β-d-glucopyranosyl adenine occurs in nine known compounds from adlay seeds, especially coixol and 2-*O*-β-glucopyranosyl-7-methoxy-2*H*-1,4-benzoxazin-3(4*H*)-one, and has shown strong melanogenesis inhibitory activity [74]. Rho-kinase inhibits the multiple biological activities of adlay seeds, and could be a scaffold for the design and development of natural Rho-kinase inhibitors [76]. 

### 3.12. Endocrine Regulation

Adlay has long been used as a treatment for dysfunctions of the endocrine system and inflammation [75]. The methanol extracts from adlay hull not only reduce testosterone release via inhibiting the protein kinase A and protein kinase C signaling pathways, 17β-HSD enzyme activity in rat Leydig cells, and GnRH-induced luteinizing hormone secretion in vitro [77], but also reduce progesterone production by inhibiting the expression of the cAMP pathway, enzyme activities, and the protein expressions of cytochrome P450 side chain cleavage enzyme and steroidogenic acute regulatory protein in rat granulosa cells [109]. Butanol extract of adlay bran not only inhibits progesterone secretion through cascade inhibitions, including the reduction in cAMP production, the PKC pathway, and the post-cAMP pathway, but also inhibits ERK1/2 protein phosphorylation before inhibiting steroidogenesis in rat granulosa cells [109].

### 3.13. Anti-Diabetes Effects

The functional ingredients of adlay with anti-diabetes effects mainly include oils, and proteins (see Table 4 and Figure 2). Adlay seed protein can effectively improve insulin resistance and inflammation in diabetic mice, and the mechanism may be through inhibition of IKK/NF-κB inflammatory pathway [79]. The excretion of bile acids and cholesterol in the feces of mice fed adlay in the diet with resistant proteins helps to inhibit plasma and liver cholesterol concentration in diabetic mice [110]. 

### 3.14. Anti-Cachexia Effects

The functional ingredients of adlay with anti-cachexia effects mainly include oils (triglycerides) (see Table 4 and Figure 2). Adlay seed oil has anti-cachexia effects by regulating the NF-κB-MuRF1 and AMPK-HSL pathways against muscle and adipose tissue loss [81]. Kanglaite is an NF-κΒ inhibitor that play an important role in many physiological and pathological processes, including cachexia, immunity, and inflammatory reactions [47].

### 3.15. Preventive of Osteoporosis 

Adlay alleviated the osteoporotic status in ovariectomized mice by increasing the proliferation of osteoblast cells via an extracellular, signal-regulated, kinase-regulated signaling pathway [13,111].

### 3.16. Analgesic Effects

Historically, adlay has been used for its anti-tumor, pain relief, anti-inflammatory, and analgesic effects. Adlay in Mauritius showed the highest fidelity (100%) for lower back pain, and was used for the first time for its potential analgesic and/or anti-nociceptive properties [112]. Adlay seed oil injection effectively reduced the pain level of cancer patients and significantly improved their quality of life [58]. Adlay seed coixol has an analgesic effect [6]. Adlay seeds (flavonoids, phytosterols, and fatty acids) can inhibit uterine contractions and relieve dysmenorrhea in vitro and in vivo [113]. Adlay has been used as therapeutic agent for dysmenorrhea as its naringenin and quercetin inhibit PGF^2α^-induced uterine contractions and inhibited uterine contraction by blocking external Ca^2+^ influx, leading to a decrease in the intracellular Ca^2+^ concentration [114]. 

### 3.17. Neuroprotection Effects

Alzheimer’s disease is characterized by the accumulation of β-amyloid in senile plaques, leading to the deterioration of brain function. Adlay hull water extracts may have a preventive therapeutic potential in the treatment of Alzheimer’s disease; particularly, its water extract Aβ25-35-induced apoptosis in dPC12 cells was associated with the enhancement of the PI3K/Akt signaling pathway [82].

### 3.18. Treatment of Gout Arthritis

Coix seed paste is effective in the treatment of gout arthritis (damp-heat bizu syndrome), with good joint recovery, pain relief, inflammatory factor reduction, no adverse reactions, and high safety [115]. Adlay seeds inhibit xanthine oxidase activity, and their extracts had hypouricemia and nephroprotective effects in hyperuricemic mice [116].

### 3.19. Extended Life

Adlay seed oil (1 mg/mL) increased the life cycle and stress resistance in *Caenorhabditis elegans* through daf-16 and its downstream genes [117].

### 3.20. Ant-Fungi Effects

A novel chymotrypsin inhibitor (12.5 mg) from adlay seeds inhibited 89% of the proteases causing spore germination in six plant-pathogenic fungi, showing a typical sequence motif of seed-stored protehistinase inhibitors [118]. Adlay seed oil exhibited anti-bacterial activity against *E. coli*, *S. aureus*, and *B. subtilis*, with a minimum inhibitory concentration of 0.031 g/L and an MBC of 0.125 g/L of oil, which was effective for all tested bacteria [119].

### 3.21. Detoxification Effects

The ethanolic root extract of adlay can effectively neutralize venom toxicity and inhibit venom-derived factors, such as LA2 activity [120]. 

In summary, both adlay seed and tartary buckwheat or barley grass and its grains have more than 20 health effects, which are supported that human has only one new cell disease theory, which are made up of sixty million cells, more than 1000 diseases are due to cell nutritional deficiencies and detoxification disorder caused by the disease [38,90]. Therefore, the functional foods of adlay seeds and barley can address these problems of human cells and oxidative stress damage, especially via their bioactive phytochemicals (e.g., favonoids, phenolic acids, triterpenoids, phytosterols saponins, polysaccharides, and alkaloids) [38].

## 4. Functional Ingredients Mechanisms in Adlay to Combat Disease 

### 4.1. Oils

Adlay oil has many bioactivities, including anti-cancer, heart disease prevention, melanogenesis inhibition, anti-cachexia, extended life, anti-fungi, and analgesic effects (Figure 1 and Figure 2, Table 3, Table 4 and Table 5). Adlay oil varies from 7.398 ± 0.486% to 8.464 ± 0.725%. Total fatty acids, triolefin, total phenolics, and total flavonoids in Xingren adlay oil were found to be the highest, the content of coixol in Pucheng adlay oil was the highest. The geographic origin has an influence on the phytochemical profiles and anti-oxidant activity of adlay oil [28]. There are 35 triglycerides, 16 diglycerides, 4 monoglycerides, 2 sterols, glycerol trioleate, fatty acids, and lipid markers in adlay seeds [121]. Thirty-two peaks in the lipid profile of adlay seeds corresponded to 20 triglycerides and 12 diglycerides, 9 of which could be used to distinguish the geographical origin [122]. Adlay bran oil has the potential to improve hyperlipidemia and hyperglycemia in diabetes by enhancing insulin sensitivity and liver glucose metabolism [123]. Coix can inhibit enzymes of cyclooxigenase, fatty acid synthase, matrix metalloproteinases, and liver cholesterol synthesis, and its extracts can inhibit NF-kB activation by inhibiting the activity of protein Kinase-C [12]. Triglycerides in adlay oil have anti-tumor effects, and Kanglaite injections with adlay oil as the main raw material have been widely used in the treatment of many cancers [6].

### 4.2. Polysaccharides

Adlay polysaccharides have many bioactivities, including anti-cancer, anti-diabetes, anti-inflammation, and gastroprotection effects (Figure 1 and Figure 2, Table 3, Table 4 and Table 5). Adlay polysaccharides have a significant inhibitory effect on the proliferation of A549 cells, inducing A549 apoptosis, and are the molecular mechanism of caseinase 3 and 9 gene expression-induced apoptosis after treatment with adlay polysaccharides [124]. Adlay polysaccharides targeting S100A4 are an alternative cancer chemotherapy, and inhibit migration and invasion of A549 cells by downregulation of S100A4, as well as possibly interacting with the binding site of S100A4-NMIIA [49]. The total polysaccharides of adlay bran not only alleviate TNF-α-evoked dysfunction of the intestinal epithelial barrier by inhibiting the NF-κB p65-mediated inflammatory reactions, but also increase paracellular permeability and reduce the transepithelial electrical resistance in TNF-α-challenged Caco-2 cells, suppress TNF-α-evoked upregulation of IL-8 and IL-6 expression, downregulate occludin and ZO-3 expression, and significantly suppress the activation and protein expression of NF-κB p65 [125]. The hypoglycemic effects of polysaccharides from adlay seeds in Type-2 diabetic mice may be related to the regulation of the intestinal microbiota and its metabolic pathways [126]. As a functional food, adlay seeds can reduce blood glucose and insulin levels [127].

### 4.3. Phenols

Adlay phenols have many bioactivities, including anti-cancer, anti-inflammation, gastroprotection, cardiovascular protection, anti-allergy, and analgesic effects (Figure 1 and Figure 2, Table 3, Table 4 and Table 5). Total polyphenols can remove free radicals, and enhance immunity, reduce blood pressure, and reduce blood sugar, and have anti-cancer, anti-inflammatory, analgesia, and anti-oxidation effects [6]. Adlay-rich diets can reduce the incidence of cancer and cardiovascular and neurodegenerative diseases. Adlay bran free phenolics are the most potent cellular anti-oxidants; they suppressed the generation of MDA, MetHb, and ROS and activated ATPase, CAT, SOD, and GSH-Px [128]. An extract (phenolics, flavonoids, coniferyl alcohol, and 4-ketopinoresinol) of adlay hull has anti-inflammatory and protective effects via Aβ-induced neurotoxicity and apoptosis in dPC12 cells, and can also treat neurodegenerative disorders (Alzheimer’s disease) [82]. The 20 phenolics (10 phenolic acids, 2 coumarins, 2 phenolic aldedhyes, and 6 flavonoids) from adlay bran, especially sinapic acid and phenolic acids, show strong xanthine oxidase inhibitory activity, and are effective drugs in the prevention and treatment of hyperuricemia [134]. Adlay tea has anti-viral effects against influenza viruses, though polyphenols might have a small effect and other ingredients might have a greater impact on the anti-viral activity [135]. Polyphenols can strengthen anti-oxidant defenses and upregulate the immune systems of COVID-19 patients, and prevent replication and spreading of the virus [136]. Ferulic acid with insoluble bounding polyphenols in adlay seeds could increase the activity of GSH-PX, CAT, and γ-GCS, and improve H2O2-induced oxidative stress in HepG2 cells via Nrf2 signaling [137].

### 4.4. Coixol

Adlay coixol has many bioactivities, including anti-cancer, anti-inflammation, melanogenesis inhibition, and analgesic effects (Figure 1 and Figure 2, Table 3, Table 4 and Table 5). The coixol of acetone extract from adlay hull can regulate gene expression by acting on airway epithelial cells [18] and the activation of five kinase signals (p38, ERK1/2, JNK, MEK1/2, and MSK1/2), and is a highly promising drug for cancer chemoprevention [17]. Coixol exhibits a strong melanogenesis inhibitory activity [74]. New lactam (coixol) from adlay bran inhibits lung and colon cancer cells with IC50 values between 28.6 and 72.6 microg/mL [59]. The effect mechanism of coixol from adlay is associated with inhibiting the activity of NF-κB, the MAPKs pathways, and the NLRP3 inflammasome [129].

### 4.5. Phytosterols

Adlay phytosterols have many bioactivities, including anti-cancer, anti-obesity, anti- diabetes, liver protection, cardiovascular protection, anti-hypertension, analgesic, and anti-allergy effects (Figure 1 and Figure 2, Table 3 and Table 4). Phytosterols are the important micronutrients in human health, with actions against inflammation, hypertension, osteoporosis, immunosuppression, metabolic abnormalities, and allergic complications. They exhibit anti-inflammatory effects via different modes through transinhibition or selective COX-2 enzymes [138]. Dehulled adlay contains phytosterols that may have played an important role in regulating plasma lipids and glucose metabolism in streptococcus-induced diabetic rats [132]. Dietary adlay oil and phytosterols reduced leptin, fatty tissue, and low-density lipoprotein cholesterol levels in rats [130]. The phytosterols (besterol, β-glusterol) in adlay can inhibit the secretion of vascular endothelial factor and its receptor activity [39]. Four steroids (campesterol, β-sitosterol, stigmasterol, and stigmas tanol) in ethyl acetate of adlay seeds at the optimal dose of 200 µg/mL inhibits the growth of endometrial cancer cells, particularly cell growth and cell cycle stagnation, most significantly in phase G1 or G2/M [41]. The phytosterols levels in an ethyl acetate fraction of adlay hull contained β-sitosterol and stigmasterol, and the efficiency of stigmasterol in terms of anti-proliferation was better than that of β-sitosterol [54]. Dehulled adlay suppressed ovalbumin/methacholine-induced acute airway inflammation, and phytosterols in adlay bran can be used as anti-allergens due to their significant anti-degranulation activities [131].

### 4.6. Resistant Starch

Adlay phytosterols have many bioactivities, including anti-cancer, anti-obesity, and anti-diabetes effects (Figure 1 and Figure 2, Table 3, Table 4 and Table 5). Adlay seed starch has a lower amylose content, with double helices and a crystalline lamellar structure and surface pinholes, and shows a higher slowly digestible starch content combined with less resistant starch fractions [126]. Amylosucrase-modified waxy adlay starch reduces short chains and increases long chains, similar to the gradual increase in resistant starch, and gradually decreases blood glucose concentrations over long periods [133]. Heat-moisture-treated adlay seeds displayed the highest resistant starch level (20.6%), facilitating molecular rearrangements and reorganization of the starch chain, forming a highly ordered molecular structure, amylose-lipids complexes, and thicker crystalline lamella [80].

## 5. Anti-Oxidants of Functional Ingredients in Adlay

### 5.1. Anti-Oxidants in Adlay

Anti-oxidants in adlay are substances that can prevent or slow damage to cells caused by free radicals, and includes oils, polyphenols, coixol, polysaccharides, phytosterols, resistant starch, flavonoids, flavones, β-carotene, vitamins (C,E), quercetin, coniferyl alcohol, syringic acid, syringaresinol, ferulic acid, 4-ketopinoresinol, mayuenolide, catechins, and selenium. Higher contents of vitamin E (8.66 mg/100 g) and flavonoids (2.26 mg/g) were found in adlay grown in Guizhou, and adlay root had the highest content of polyphenols (4.53 mg/g) [14].

### 5.2. Anti-Oxidants to Prevent Chronic Diseases

Adlay oil has many bioactivities, including anti-cancer, heart disease prevention, melanogenesis inhibition, anti-cachexia, extended life, anti-fungi, and analgesic effects Adlay phenols also have many bioactivities, including anti-cancer, anti-inflammation, gastroprotection, cardiovascular protection, anti-allergy, and analgesic effects. Adlay oil components with anti-oxidant activity include total fatty acids, triolein, total phenolics, and total flavonoids, especially 2,2′-azino-bis (3-ethylbenzothiazoline-6-sulfonic acid), with its powerful radical scavenging activity (IC50 from 0.924 to 1.116 mg/mL), ferric- reducing anti-oxidant power activity (EC50 from 0.019 to 0.028 mg/mL), and β-carotene-linoleic acid (IC50 from 0.233 to 0.414 mg/mL) [28]. Lipophilic extract from fermented coix seed exhibited higher anti-oxidant activity in scavenging free radicals and inhibiting lipid oxidation [53]. Germination of adlay seeds significantly increased the free form phenolic and flavonoid contents by 112.5% and 168.3%, respectively, which led to a significant enhancement of the anti-oxidant activities [139]. The free, bound, and total flavonoid contents of adlay range from 6.21–18.24, 18.68–35.27, and 24.88–52.86 mg, respectively, of catechin equiv/100 g. Total polyphenol and anti-oxidant capacity of Liaoning 5 adlay and Longyi 1 adlay are significantly better than Guizhou heigu adlay [140]. A new phenolic compound, N^1^,N^5^-di-[(*E*)-*p*-coumaroyl]-spermidine from adlay, could protect HepG2 cells from oxidative stress by increasing the anti-oxidant enzymes regulated by the Nrf2/ARE pathway [141]. Vitamin E in adlay prevents chronic diseases [70].

Polysaccharides have anti-oxidant, anti-virus, and immunity improving effects [142]. Adlay polysaccharides have many bioactivities, including anti-cancer, anti-diabetes, anti-inflammation, and gastroprotection effects. Polysaccharides have significant anti-oxidant effects via the endogenous anti-oxidant stress Nrf2/ARE pathway, regulating the expression of downstream anti-oxidant enzymes, especially their multi-channel, multi-target, multi-effect, and other characteristics [142]. Phytosterols have anti-oxidant, anti-inflammatory, cholesterol-lowering, and other biological activities [143]. Adlay phytosterols have many bioactivities, including anti-cancer, anti-obesity, anti-diabetes, liver protection, cardiovascular protection, anti-hypertension, analgesic, and anti-allergy effects. Resistant starch has a certain anti-oxidant ability, especially black bran [144].

## 6. The Food and Pharmaceutical Industry

### 6.1. Food Industry

Adlay in food and fermentation has broad prospects and a long history, such as in beer brewing recipes that are over 5000 years old, lipophilic nutraceuticals, fermented foods, and functional foods. Pottery vessels from the Mijiaya site revealed a surprising beer recipe fermented about 5000 years ago with broom corn millet, barley, adlay seed tears, and tubers [145]. Seyoeum (adlay, rice, sesame, soybeans, Chinese yam, and *Liriope platyphylla*) may be a functional meal replacement with anti-obesity and anti-insulin resistance effects [146]. The fusion of nanoparticles with the membrane enabled the transport of adlay seed oil extract, improved the intestinal permeability of the Caco-2 cell monolayer, and supported the design of homologous polysaccharide and protein-based delivery systems to increase the bioavailability of lipophilic functional foods [147].

Adlay seed fermentation with *L. plantarum* NCU137 could improve its nutritional, sensory, and stability properties [148]. Adlay seeds and food should be protected against fungal infection; 89 genera and 96 species of fungi have been found on coix seeds, especially Ascomycota (81.48%), Basidiomycota (4.08%), Xeromyces (8.50%), Gibberella (7.25%), and Aspergillus (4.74%) [149]. The aging mechanism of coix seeds during storage includes starch and sucrose metabolism, carbon metabolism, RNA transport, proteasome, protein processing in the endoplasmic reticulum, ribosome, RNA degradation, and upregulated proteins of anti-oxidants and resistance stimulus [150]. The pathway of flavonoid biosynthesis, starch and sucrose metabolism, lipid metabolism, and amino acid metabolism are related to the nutritional quality of adlay seeds, and their molecular mechanisms of key genes and proteins are associated with the metabolism and accumulation of nutrients [8]. Fructooligosaccharide from adlay seeds, obtained by hot water extraction at 60 °C for 1 h, can used as an anti-oxidant source in food or cosmetic products [151]. The combined systems of adlay seeds oil and β-carotene exhibited higher bioavailability, alongside increased anti-cancer and anti-oxidant activity, which means they could be used for functional foods and supplements [152]. Butyryl galactose ester-modified coix component micro-emulsions (50 μg/mL) treatment resulted in a 1.34-fold rise in total apoptosis of cells of HepG2, and was developed for enhanced liver tumor-specific targeting and as a novel ligand for realizing hepatoma-targeting drugs delivery [153]. A diet with dehulled adlay can ameliorate non-alcoholic fatty liver disease progression by decreasing insulin resistance, steatosis, and inflammation [154].

### 6.2. Kanglaite Injection of Anti-Cancer Drugs

Adlay in the pharmaceutical industry has broad prospects of application. Drugs with adlay oil have been applied to treat multiple cancers. Kanglaite injection, a commercial product of adlay seed oil, has been used clinically as an anti-cancer drug in China for decades [52], and in many health food and medicinal diets [6]. Dodecanoic acid, tetradecanoic acid, 2,3-dihydroxypropylester, 1,3-dioctanoin, *N*-methoxy-*N*-methyl -3,4-dihydro-2*H*-thiopyran6-carboxamide, 5-Amino-1-(quinolin-8-yl)-1,2,3-triazole-4- carboxamide, propanamide, and pyridine have been identified in adlay seeds oil [119]. Kanglaite injection from adlay seeds extracted is an NF-κΒ inhibitor that has been widely used for gastric cancer and advanced lung cancer [155,156]. Kanglaite pretreatment can not only increase the effect of taxol on colorectal cancer and cisplatin on HepG2 cells by inhibiting the CKLF1-mediated NF-κB pathway [157,158], but also inhibits tumor necrosis factor-alpha-mediated epithelial mesenchymal transition in colorectal cancer cell lines by inhibiting NF-κΒ [47]. However, Kanglaite can induce or inhibit the activities of cytochrome P450, which may lead to herb–drug interactions [159]. Adlay seed oil not only regulates the triple-negative breast cancer metabolism through the miR-205/S1PR1(sheath osine 1 phosphate receptor 1) axis and the downstream STAT3/MAPK/AKT signaling pathway, but also reduces the expression of S1PR1, cyclinD1, and phosphorylation levels in STAT3, MAPK, and AKT, while upregulating p27 [52]. Kanglaite^®^ injection enhances the efficacy and reduces the side effects of chemotherapy, improving the quality of life in patients with gastric cancer, and significantly reducing the rate of gastrointestinal reactions and bone marrow suppression [160].

### 6.3. Other Drugs

Adlay seeds are the most common functional food crop in Chinese medicine, especially at the Hemudu site and during the Xia Dynasty [6]. Adlay Fuzi Baijiangsan can upregulate Nrf2 expression, especially its downstream anti-oxidant protein HO-1, and increase Nrf2 mRNA expression in ulcerative colitis [161]. An adlay seed emulsion abrogated gemcitabine-induced activation of NF-κB, and synergistically sensitized pancreatic cancer cell therapy [44]. The combination of norcantharidin + adlay seed oil improves the anti-tumor activity and modulates tumor-infiltrating Tregs [162]. Dehulled adlay fermented with *B. subtilis* may be a potential therapeutic agent for leukemia and lung cancer [163]. Delayed diets with a novel adlay seed extrusion cooked-based synbiotic ameliorates high-fat diet-induced obesity, metabolic disorders, and dysbiosis [164]. Linoleic acid (28.59%), oleic acid (56.95%), 3-*O*-(*trans-*4-feruloyl)-β-sitostanol (0.15%), 3-*O*-(*cis*-4-feruloyl)-β-sitostanol (0.02%), and β-sitosterol (0.41%) in adlay bran are upregulate in lipoprotein lipase, AMPK, and p-AMPK, and downregulated in fatty acid synthase, making adlay a promising functional food or new botanical drug [165]. In addition, the bioavailability and application of flavonoids can support the formulation of new natural food additives with beneficial health effects [166].

Adlay and its extracts are rich in 30 ingredients with more than 20 health effects. The effect of adlay on human health has been confirmed by PubMed and Google search, including its anti-cancer [46,167], anti-inflammation [97], anti-obesity [11,12,97], liver protection [140], anti-virus [8,105,106], cardiovascular protection [11], anti-hypertension [11,108], heart disease prevention [11,70], endocrine regulation [75], anti-diabetes [132], osteoporosis prevention [12], and analgesic effects [113].

## 7. Conclusions and Future Perspectives

There are 30 functional components in adlay’s extracts, and six parts (adlay, coix coat, coix hull, coix root, coix stem, and coix leaves) of *Coix lacryma jobi*, which can be used as Chinese herbal medicine. The adlay, roots, stems, and leaves have been widely used in functional food and medicine, and are known as “the King of Gramineae”. Adlay and *Coix* leaves are the best ideal raw materials for human functional food and livestock functional forages, based on the functional ingredients in adlay (eight) and Coix leaves (nine), being at the highest level of the six tissues. This review paper summarizes the 20 health effects of adlay based on human and animal and cell culture experiments, although nearly half of the health effects need to be further verified in human clinical trials. It is worth noting that the functional ingredients of adlay have been used in the treatment of thirteen types of cancer. The anti-cancer mechanism is relatively complex; the anti-tumor mechanism may involve multiple components, channels, and targets. Further research is also needed to fully clarify the anti-tumor mechanism of adlay seeds.

Six tissues (adlay, coat, hull, root, stem, and leaves) of *Coix lacryma-jobi* L. var. *ma-yuen* Stapf are rich in 30 ingredients, and have a food structure enabling them to demonstrate more than 20 health effects, especially the molecular mechanisms and anti-oxidants of six functional ingredients in adlay (oils, polysaccharides, phenols, coixol, phytosterols, and resistant starch), which have 10 major health effects. These results suggest that barley plays an important role in a healthy diet, and the early humans overcame a variety of chronic diseases. Adlay in the pharmaceutical industry has broad prospects of application. For example, Coix oil is used as a raw material for Kanglaite injections, and in many health foods and medicinal diets. Function components with anti-oxidants are rich in adlay. The health effect of functional components in adlay is the result of the long-term continuous evolution of early hominids and *Homo sapiens* alongside functional foods.

Although understanding the pharmacological action mechanisms of functional ingredients in *Coix lacryma-jobi* L. var. *ma-yuen* Stapf for the prevention of human chronic diseases is a very complicated task, it is essential to develop new drugs and new functional foods to prevent and control human chronic diseases. It is necessary to conduct more systemic studies to unravel the coevolutionary interconnection mechanisms existing between chronic diseases and the human diet. Unfortunately, so far there is no evidence of adlay evolving during evolutionary consumption to combat chronic diseases. Adlay seed composition is complex, and some component structures have not been studied. Adlay plays an important role in promoting the development of new drugs and functional foods and has potential underlying molecular mechanisms and action mechanisms which are worthy of further study. Other parts of adlay, including the bran, hull, root, stem, and leaves, have effective ingredients, and their contents are not less than the content of the seed. Whether these parts can be used as medicinal sites also needs further research. This review may be used as a starting point for the development of new drugs and novel functional foods from Coix to improve the prognosis of chronic diseases.

## Figures and Tables

**Figure 1 molecules-27-04808-f001:**
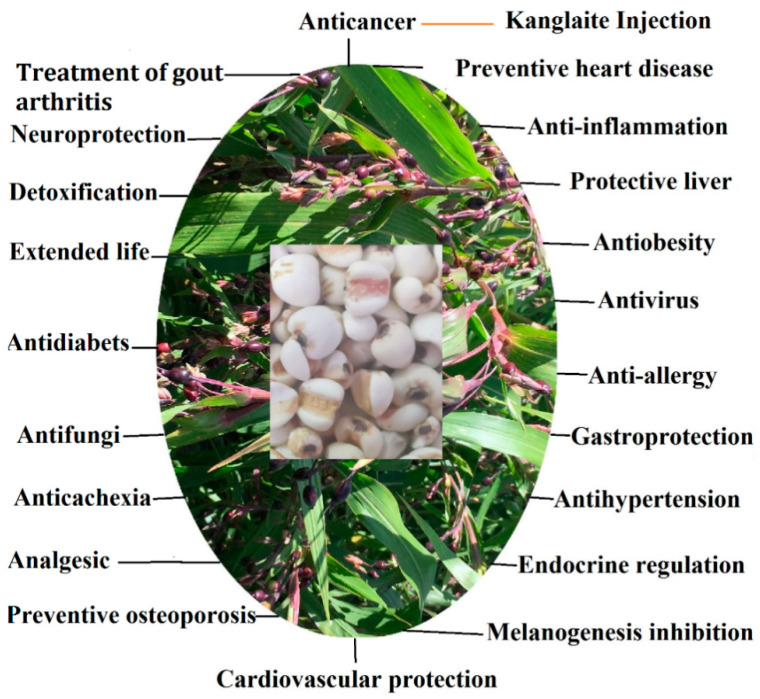
Functional ingredients of adlay combat more than 20 chronic diseases.

**Figure 2 molecules-27-04808-f002:**
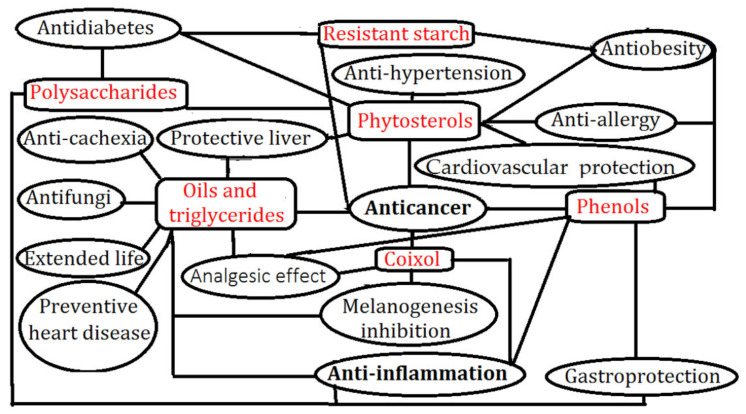
Actional relationship between functional components of adlay and chronic disease.

**Table 1 molecules-27-04808-t001:** Nutrition and functional ingredients in different tissues of adlay [14].

Components	Adlay Seeds	Adlay Coat	Adlay Hull	Adlay Root	Adlay Stem	Adlay Leaves
Total Ash (%)	2.65	15.68	21.32	5.89	1.83	14.70
Protein (%)	19.33	10.85	7.56	19.06	12.07	18.29
Crude fiber (%)	2.05	53.56	45.43	28.78	43.44	18.95
Crude fat (%)	4.70	5.31	4.52	7.15	2.44	0.38
Polysaccharides (%)	2.26	0.65	0.52	0.53	0.28	0.47
Starch (%)	48.58	3.55	4.56	6.78	8.89	3.21
VE (mg/kg)	86.61	106.85	119.27	146.5	109.32	112.36
VC (mg/kg)	0.11	5 0.08	0.12	8 0.26	0.28	0.31
VB1 (mg/kg)	0.51	0.56	0.42	0.61	0.44	0.58
VB2 (mg/kg)	0.62	0.78	1.28	2.96	1.66	0.99
VB6 (mg/kg)	8.38	9.09	12.66	21.64	7.72	40.07
VB12 (mg/kg)	0.54	0.66	—	1.32	0.43	0.89
Folciate acid (mg/kg)	0.12	—	0.05	2.51	1.97	8.85
Nicic acid (mg/kg)	12.36	14.97	13.21	4.92	—	3.27
K (mg/kg)	2165.50	2294.61	1987.05	3714.79	2468.96	4175.83
Ca (mg/kg)	305.39	609.51	875.69	1845.33	1374.30	15,645.02
Na (mg/kg)	1020.42	1231.67	1393.59	877.92	1060.83	1904.38
Mg (mg/kg)	1212.29	1481.48	1567.37	2385.63	2283.13	5845.83
Zn (mg/kg)	146.83	88.93	91.36	84.01	20.91	41.22
Fe (mg/kg)	116.23	107.89	90.47	1459.28	98.92	367.18
P (mg/kg)	1933.20	323.93	318.92	849.51	391.89	1405.92
Se (mg/kg)	0.04	0.09	0.07	0.11	—	0.04
Total amino acids (%)	14.400	3.804	1.654	6.612	3.394	8.278
Palmitate * (%)	0.68	8.42	10.52	8.42	9.53	7.93
Palm-linoleic acid * (%)	0.59	—	—	1.36	—	—
Stearic acid * (%)	5.64	2.35	1.96	3.35	2.12	2.93
Oleic acid * (%)	40.59	43.55	21.17	26.74	43.87	46.11
Linoleic acid * (%)	26.53	31.18	14.49	18.78	29.88	33.47
Decoanic acid * (%)	1.10	1.04	0.90	0.75	0.87	1.24
Eoixol (mg/g)	0.14	0.25	0.23	1.26	0.34	0.34
Flavonoids (mg/g)	2.26	0.13	0.09	0.21	0.44	2.13
Total phenol (mg/g)	0.27	1.89	1.04	4.53	3.07	3.15

Note: * Refers to the percentage of six monomeric fatty acids in the total fatty acids.

**Table 2 molecules-27-04808-t002:** Nutrition and functional ingredients of adlay seeds.

Composition	Mean ± SD	Range	References
Polysaccharides (%)	1.59 ± 0.95	0.92–2.26	[14,26]
Total starch (%)	65.54 ± 3.28	57.82–71.51	[21]
Amylase (%)	4.87 ± 5.69	0.00–25.48	[21]
Resistant starch (%)	5.57 ± 0.08	5.49–5.65	[27]
Lipid (fat+oil) (%)	7.82 ± 0.60	6.32–9.13	[21]
Triolein (%)	0.82 ± 0.19	0.53–1.04	[22]
Protein (%)	18.18 ± 1.36	14.77–21.78	[21]
Total amino acid (%)	17.38 ± 1.21	14.02–20.67	[21]
Flavonoids (mg/g)	1.24 ± 1.44	0.22–2.26	[14,28]
Polyphenols (mg/g)	0.81 ± 0.76	0.27–1.35	[14]
Eoixol (mg/g)	29.92 ± 42.12	0.14–59.70	[14,24]
γ-oryzanol (μg/g)	271.64 ± 101.36	176.52–375.35	[29]
γ-Tocopherol (μg/g)	29.72 ± 10.34	22.41–37.03	[29]

Note: Total starch includes amylose and resistant starch, lipids include fats and oils as well as triolein, and protein includes total amino acids.

**Table 3 molecules-27-04808-t003:** Action pathway and functional ingredients of adlay in the treatment 13 cancer types.

Pharmaco- Logical Action	Anti- Cancer Type	Action Pathway	Functional Ingredients (Study Types)	References
Inhibiting tumor micro- angiogenesis	Cervical cancer	Inhibiting vascular endothelial factor secretion and its receptor activity	Besterol, β-glusterol (human clinical trials)	[6,39]
Induced the apoptosis of cancer cells	Cervical cancers	Inactivation of the PI3K/AKT pathway in HeLa cells inhibited tumor cell proliferation, enhanced anti-angiogenesis, and induced apoptosis by regulating bax/ bcl-2 and the activating caspase-3 pathway.	Transferrin-functionalized microemulsions coloaded; Coix seed oil (human clinical trials)	[35,40]
	Hepatoma cancer	Regulating the expression of caspase-8; the half- maximal inhibitory effect against HepG2 cells was 46.5 ± 2.4μg/mL	Octanoyl galactose ester; adlay extract (in vitro, in vivo)	[41,42]
	Pancreatic Cancer	Downregulation of BCL-2 protein expression, increased expression of the Fas gene, apoptosis by activating caspase-3 and increasing the Bax/Bcl-2 ratio, and inhibition of NF-κB activity and downstream target genes.	Triglycerides, Coixenolide, Coixol, coix seed emulsion (human clinical trials)	[6,43,44]
	Early mature leukemia	Activated apoptotic protein Caspase-3, induced mitochondrial apoptosis, and promoted cell apoptosis.	Coixenolide (human clinical trials)	[45]
	Non-small cell lung cancer	Activating the endogenous mitochondrial apoptosis pathway and activating the apoptotic proteins Caspase-6 and Caspase-9 and promotes apoptosis.	Polysaccharides (human clinical trials)	[6,46]
Inhibiting cancer cell metastasis	Colorectal cancer	Inhibits the tumor necrosis factor-α-mediated epithelial mesenchymal transition via the inhibition of NF-κΒ.	Resistant starch, coix seed oil (human clinical trials)	[38,47]
	Colon cancer	HUVECs via repression of the ERK1/2 and AKT pathways.	Adlay sprout extract (human clinical trials)	[48]
	Non-small cell lung cancer	Downregulation of S100 calcium-binding protein and A4 expression and supports the expression of proliferative proteins, invasive proteins, matrix proteins, and the JAK2/STAT3 signaling pathway.	Polysaccharides, Triglycerides, Coixenolide, Coixol (human clinical trials)	[6,49,50]
	Laryngeal cancer	Inhibits the expression of invasion and transfer factors.	Triglycerides, Coixenolide, Coixol (human clinical trials)	[51]
Suppress the cancer cell proliferation	Breast cancer	miR-205/S1PR1 regulate sphingomyelin metabolism, downstream STAT3/MAPK/AKT signal pathways.	Coix seed oil (animal study)	[52]
	Endometr- ial cancer	Ethyl acetate showed dose-dependent cell cycle arrest of HEC-1A and RL95-2 cells at the sub G1 checkpoint and G2/M checkpoint.	Polyphenols, Flavonoids, Phytosterols, Fatty acid (in vitro)	[41]
	Laryngeal cancer	Meddling with protein phosphatase 2A in the sphingom-yelin cycle and blocking the cell cycle at the G0/G1 phase.	γ-oryzanol, Coixenolide (in vitro)	[53]
	Myterine oma	Inhibits ustrosphyperplasia induced by sex hormone hexene estrogol/methoxyprogesterone 17-acetate.	Besterol, β-glusterol (animal study)	[6,54]
	Gastric cancer	Inhibits Bcl-2 gene expression and blocks cancer cells in the G1 cycle.	Triglycerides, Coixenolide, Coixol (human clinical trials)	[6,55]
	Lung cancer	Suppresses cytocyclin A expression and blocks cancer cell proliferation during the cell cycle G1/S transition.	Adlay extract, Triterpene (in vitro, in vivo)	[56,57]
	Uterine leiomyoma	Reduced diethylstilbestrol/medroxyprogesterone 17-acetate-induced uterine myometrial hyperplasia.	Stigmasterol, β-sitosterol (animal study)	[54]
Anti-cachexia	Lewis lung cancer	Adjusts the NF-κB-MuRF1 and AMPK-HSL pathways to inhibit elevated inflammatory factors and leads to phosphorylation of HSL to avoid fat and muscle loss.	Coix seed oil (human clinical trials)	[58,59]

**Table 4 molecules-27-04808-t004:** Action pathway and functional ingredients of adlay in preventing chronic diseases other than cancers.

Chronic Diseases	Action Pathway	Functional Ingredients	References
Anti- inflammation	Cell membrane into the cytosol and nucleus, triggering gene expression, changes in cell proliferation, and the induction of apoptosis; increased cellular production of nitric oxide and prostaglandin E2 by downregulating nitric oxide synthase and cyclooxygenase 2 expression.	Polyphenols, coixol, polysaccharides, eriodictyol, ceramide, *p*-coumaric acid	[6,60,61]
Anti-obesity	Glycerolipid metabolism, biosynthesis of unsaturated fatty acids, sulfur reduction, and glutathione transport system; AMP-activated protein kinase in adipose differentiation decrease adipogenesis expression and enhances binding protein α.	Polyphenols, dietary fiber, adlay extract, phytosterols, resistant starch	[38,62,63]
Protective liver	Inhibiting vascular endothelial factor secretion and its receptor activity.	Besterol, β-glusterol	[6,39]
Anti-virus	Increase in peripheral cytotoxic lymphocytes and enhanced cytotoxic activity.	Adlay extract	[64]
Cardiovascular protection	Reduced serum cholesterol and low-density lipoprotein cholesterol and improved high-density lipoprotein cholesterol.	Polyphenol	[37,65]
Gastro- protection	Reducing the pH value of gastric juice, increasing the density and length of gastrointestinal villi, and modulating gut microbiota.	Phenolic acids, caffeic, chlorogenic acids, polysaccharides	[66]
Anti- hypertension	Angiotensin converting enzyme inhibitory peptide GAAGGAF by glutelin and polypeptides produced by gluten; high angiotensin I conversion of enzyme inhibitory activity tetrapeptide downregulated expression.	Glutelin, gluten	[6,67,68]
Preventive heart disease	Assembled gene sets by the levels of prolamin and vitamin E biosynthesis-associated protein; blood lipid-reducing and anti-oxidant effects.	Vitamin E, adlay seed oil	[69,70]
Anti-allergy	Inhibit mast cell degranulation, suppress the production of Akt, influence the signal transduction in cells; inhibitory effect on allergic response via the ERK signaling transduction in RBL-2H3 cells.	Phenolic acids; hydroxyacetophenone, flavone *p*-coumaric acid	[71,72]
Melanogenesis inhibition	Reduce tyrosinase activity and melanin production in melanoma cells, which has a tyrosinase inhibitor and anti-hyperpigmentation agent.	Adlay bran oil, coixol	[73,74]
Endocrine regulation	Decrease testosterone release via the inhibition of the protein kinase A/C signal transduction pathways; GnRH-induced luteinizing hormone secretion; inhibited progesterone secretion and reduced cAMP, PKC, and post-cAMP pathway action.	Methanol extracts of adlay hull	[75,76,77]
Anti-diabetes	Improve insulin resistance by inhibition of the IKK/NF-κB inflammatory pathway; regulation of the intestinal microbiota and its metabolic pathways.	Protein, resistant starch; polysaccharides phytosterols	[38,78,79,80]
Anti-cachexia	Counteract muscle and adipose tissue loss through regulating the NF-κB-MuRF1 and AMPK-HSL pathways.	Coix seed oil	[81]
Preventive osteoporosis	Increase the proliferation of osteoblast cells via an extracellular signal-regulated kinase-regulated signaling pathway.	Water extract of adlay	[13,79]
Neuroprotect- ion	Aβ25-35-induced apoptosis in dPC12 cells was associated with the enhancement of the PI3K/Akt signaling pathway	Water extract of adlay hull	[82]

**Table 5 molecules-27-04808-t005:** Relationship between functional components of adlay and chronic disease.

Functional Ingredients	Chronic Diseases	Efficacy Number
Oils and triglycerides	Anti-cancer [6,40,85,92], anti-inflammation [62], liver protection [101], preventive heart disease [69], melanogenesis inhibition [73], anti-cachexia [81], analgesic [6], extended life [118], anti-fungi [119]	9
Polysaccharides	Anti-cancer [49,60,124], anti-inflammation [61,125], gastroprotection [126], anti-diabetes [127]	4
Phenols	Anti-cancer [6,41], anti-inflammation [6,96], anti-obesity [38], cardiovascular protection [37,128], gastroprotection [108], anti-allergy [71], analgesic [113]	7
Coixol	Anti-cancer [17,44,59], anti-inflammation [129], melanogenesis inhibition [74], analgesic [5]	4
Phytosterols	Anti-cancer [41,54], anti-obesity [130], protective liver [100,101], cardiovascular protection [39,41], anti-hypertension [67,68], anti-allergy [131], anti-diabetes [132], analgesic [113]	8
Resistant starch	Anti-cancer [47], anti-obesity [62], anti-diabetes [126,133]	3

## Data Availability

Not applicable.

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
