# Peer review of "Actional Mechanisms of Active Ingredients in Functional Food Adlay for Human Health"

_molecules, 2022, doi:10.3390/molecules27154808_

Round 1
Reviewer 1 Report
This review summarizes the preventive effects of functional components of Coix Seed on different diseases, and integrates the pharmacological effects that Coix Seed may affect and the regulatory mechanisms involved at the efficacy level. I suggest that this review can include two more articles, the first is Endocrine Regulation in Chapter 3.12, which can be added PMID: 16254570, and the second is Chapter 3.16 Analgesia, which can be added PMID: 18577689. In addition, I suggest redrawing Figures 1 and 2, the figures should be clearer and less complex to be easy to read. If the revision can be completed, I can recommend the manuscript for acceptance.
Reviewer 2 Report
This review deals with the composition of adlay, the bioactivity of its components and their mechanisms of action. Although the subject seems interesting and the review apparently is very completed, I think before it can be considered for publication it needs a thoroughly revision of the English language, some parts of the paper are a collection of non-sense sentences.
Only Some examples, there are much more along the text:
Line 34 to 35: “Human health is dependent on a plentiful supply of functional foods [1], especially functional rice, barley, adlay and so on.” Why specially rice, barley and adlay? What that it means “so on”?.
Line 40 to 41: “Its roots, stems, and leaves also have medicinal and edible, known as “the King of Gramineae” [3]” What is the meaning of this sentence?
Line 41 to 43: “Its secondary diversity center located the hilly region of South China, especially Guizhou showed the largest genetic diversity, the molecular variation within populations was much higher than that of geographical regions [4].” What is a secondary diversity center? What does this sentence means?.
Line 46 to 48: “Coix comprises 9-11 species with different ploidy levels, which reveal diploidization of Coix lacryma-jobi (2n = 20) during evolution and Coix aquatica of Hybrid Guangxi is a recently formed hybrid [6].” Please revise this sentence.
Line 77 to 82: What does it means that eight, eleven and nine functional ingredients contents in adlay seeds, roots and leaves are the highest in six tissues?
Line 111 to 114: “The biotransformation of high-yield tetramethylpyrazine (6.93mg/g), fibrinolytic enzyme (2236.17 U/g) and functional ingredients (phenolics, GABA, triterpenes, flavonoids and coixenolide) of fermented adlay products,which is more suitable for human consumption[30]” What does this means?.
Line 118 to 120: “The hexane soluble fraction of the Laos adlay roasted hull showed the highest linoleic acid content(8.09± 0.74%), however all ethyl acetate fractions with high carotenoid and tannin[32]” What does this means?.
Line 131: “This review is mainly to combat more than 20 chronic diseases in human being:” Unfortunately a review can not combat any disease.
Line 136 to 142: It is unclear the idea that the authors want to develop in this paragraph.
Please avoid to finish every list of something wit “so on”.
Please avoid long list of illnesses.
Reviewer 3 Report
In this manuscript, Zheng et al., reviewed potential effects, mechanisms of action and active ingredients of a medicinal/food homologous adlay on many human chronic diseases.
Comments:
1. Are there any epidemiological evidence (human trials) and systematic reviews to use adlay? If so, these information should be presented in the Introduction.
2. Table 2. There were great differences in contents of starch (65.54 ± 3.28 %), fat (7.82 ± 0.60%), protein (18.18 ± 1.36%) and amino acid (17.38 ± 1.21%) among 86 adlay landraces [21]. If you add up these numbers are over 100%. The same question for the % in Table 1. It seems they are over 100%.
3. “This review is mainly to combat more than 20 chronic diseases”. I suggest you only focus on 4-5 chronic diseases with clear evidence from human studies. Many claims based purely on a few cell cultures should be removed. Section 3.3. Anti-obesity. It seems no human studies was mentioned. How much Adlay (or adlay extracts) should be consumed to have an anti-obesity effect (lose weight) in humans?
4. Table 3. “Action pathway and functional ingredients of Adlay in treated thirteen cancers”. You need to indicate the type of the study, in vitro or in vivo, animal study or human clinical trials etc.
5. The conclusion section should be concise. I suggest the authors to remove repeated background information from this section.
6. A proof-reading by a native English speaker is needed. A search found there are 22 “so on” were used in the text.
Round 2
Reviewer 2 Report
The paper has been clearly improved in the revised version. Only some minor observations:
-Line 41: Talking about Blueberry seems out of context.
-Line 91: I don´t understand the meaning of the sentence "Eight functional ingredients (...) in adlay seeds were at the highest level in six tissues (.....). Do you mean "....at the higest level among six tissues..."?. The same for the two following sentences.
-Line 99: Probably you mean "These findings show that....".
-Line 414 to 416: Please revise the sentence. What does it means "that human has only one new cell disease theory" ?
Author Response
R2: The paper has been clearly improved in the revised version. Only some minor observations:
- Line 41: Talking about Blueberry seems out of context.
Reply: Thanks so much for your advice; Use the following references instead: Functional rice and barley have become an important part of functional food crops, while functional food crops lead the development of functional agriculture with plateau characteristics [3].
- Line 91: I don´t understand the meaning of the sentence "Eight functional ingredients (...) in adlay seeds were at the highest level in six tissues (.....). Do you mean "....at the higest level among six tissues..."?. The same for the two following sentences.
Reply: Thanks so much for your advice; Eight functional ingredients (protein, polysaccharides, resistant starch, Zn, P, amino acids, palmitate, and flavonoids) in adlay seeds were at the highest level among six tissues (seeds, coat, hull, root, stem, leaves) of adlay (see Table 1).=Protein content in adlay seeds were at the highest level among six tissues (seeds, coat, hull, ro ot, stem, leaves) of adlay; Polysaccharides content in adlay seeds were at the highest level among six tissues of adlay; Resistant starch content in adlay seeds were at the highest level among six tissues of adlay; Zn content in adlay seeds were at the highest level among six tissues of adlay; P content in adlay seeds were at the highest level among six tissues of adlay;Amino acids content in adlay seeds were at the highest level among six tissuesof adlay;Palmitate content in adlay seeds were at the highest level among six tissues of adlay;Flavonoids content in adlay seeds were at the highest level among six tissues of adlay.
Eleven functional ingredient contents (crude fat, five vitamins, Fe, Se, palm-linoleic acid, eoixol, total phenol) contents in adlay root were at the highest level among six tissues (seeds, coat, hull, root, stem, leaves) of adlay (see Table 1).= Crude fat content in adlay root were at the highest level among six tissues (seeds, coat, hull, root, stem, leaves) of adlay; Five vitamins content in adlay root were at the highest level among six tissues of adlay; Fe content in adlay root were at the highest level among six tissues of adlay; Se content in adlay r oot were at the highest level among six tissues (seeds, coat, hull, root, stem, leaves) of adlay; Palm-linoleic acid content in adlay root were at the highest level among six tissues of adlay; Coixol content in adlay root were at the highest level among six tissues of adlay; Total phenol content in adlay root were at the highest level among six tissues of adlay.
Nine functional ingredient contents (VB6, folciate acid, K, Ca, Na, Mg, oleic acid, linoleic acid, and decoanic acid) in adlay leaves had the highest levels among six tissues (seeds, coat, hull, root, stem, leaves) of adlay (see Table 1)= VB6 content in adlay leaves had the highest levels among six tissues (seeds, coat, hull, root, stem, leaves) of adlay; Folciate acid content in adlay leaves had the highest levels among six tissues of adlay; K content in adlay leaves had the highest levels among six tissues of adlay; Ca content in adlay leaves had the highest levels among six tissues of adlay; Na content in adlay leaves had the highest levels among six tissues of adlay; Mg content in adlay leaves had the highest levels among six tissues of adlay; Oleic acid content in adlay leaves had the highest levels among six tissues of adlay; Linoleic acid content in adlay leaves had the highest levels among six tissues of adlay; decoanic acid content in adlay leaves had the highest levels among six tissues of adlay.
- Line 99: Probably you mean "These findings show that....".
Reply: Thank you very much for your suggestions that have been modified.
4. Line 414 to 416: Please revise the sentence. What does it means "that human has only one new cell disease theory" ?
Thank you very much for your suggestions that have been modified. “that human has only one new cell disease theory,which are made up of sixty million cells, more than 1,000 diseases are due to cell nutritional deficiencies and detoxification disorder caused by the disease [38,53].”

Reviewer 3 Report
This revised version has been improved. I can recommend to accept this version for publication.
Author Response
The modified part of molecules-1744999 uses the red font standard.
